# Does One Size Fit All? Variations in the DNA Barcode Gaps of Macrofungal Genera

**DOI:** 10.3390/jof9080788

**Published:** 2023-07-26

**Authors:** Andrew W. Wilson, Ursula Eberhardt, Nhu Nguyen, Chance R. Noffsinger, Rachel A. Swenie, Justin L. Loucks, Brian A. Perry, Mariana Herrera, Todd W. Osmundson, Sarah DeLong-Duhon, Henry J. Beker, Gregory M. Mueller

**Affiliations:** 1Denver Botanic Gardens, 909 York Street, Denver, CO 80206, USA; 2Staatliches Museum für Naturkunde Stuttgart, Rosenstein 1, 70191 Stuttgart, Germany; 3Department of Tropical Plant and Soil Sciences, University of Hawaiʻi at Mānoa, 3190 Maile Way, St. John 102, Honolulu, HI 96822, USA; 4Department of Ecology and Evolutionary Biology, University of Tennessee, Dabney Hall, 1416 Circle Drive, Knoxville, TN 37996, USA; 5Department of Biological Sciences, California State University East Bay, 25800 Carlos Bee Blvd., Hayward, CA 94542, USA; 6Chicago Botanic Garden, 1000 Lake Cook Road, Glencoe, IL 60022, USA; 7Biology Department, University of Wisconsin-La Crosse, 1725 State Street, La Crosse, WI 54601, USA; 8Department of Biology, The University of Iowa, Iowa City, IA 52245, USA; 9Royal Holloway College, University of London, London WC1E 7HU, UK; 10Plantentuin Meise, Nieuwelaan 38, B-1860 Meise, Belgium

**Keywords:** DNA barcode, pairwise distances, nrITS, ITS1, ITS2

## Abstract

The nuclear ribosomal internal transcribed spacer (nrITS) region has been widely used in fungal diversity studies. Environmental metabarcoding has increased the importance of the fungal DNA barcode in documenting fungal diversity and distribution. The DNA barcode gap is seen as the difference between intra- and inter-specific pairwise distances in a DNA barcode. The current understanding of the barcode gap in macrofungi is limited, inhibiting the development of best practices in applying the nrITS region toward research on fungal diversity. This study examined the barcode gap using 5146 sequences representing 717 species of macrofungi from eleven genera, eight orders and two phyla in datasets assembled by taxonomic experts. Intra- and inter-specific pairwise distances were measured from sequence and phylogenetic data. The results demonstrate that barcode gaps are influenced by differences in intra- and inter-specific variance in pairwise distances. In terms of DNA barcode behavior, variance is greater in the ITS1 than ITS2, and variance is greater in both relative to the combined nrITS region. Due to the difference in variance, the barcode gaps in the ITS2 region are greater than in the ITS1. Additionally, the taxonomic approach of “splitting” taxa into numerous taxonomic units produces greater barcode gaps when compared to “lumping”. The results show variability in the barcode gaps between fungal taxa, demonstrating a need to understand the accuracy of DNA barcoding in quantifying species richness. For taxonomic studies, variability in nrITS sequence data supports the application of multiple molecular markers to corroborate the taxonomic and systematic delineation of species.

## 1. Introduction

The nuclear ribosomal internal transcribed spacer (nrITS) region is the DNA barcode for fungi [1]. Since its establishment as the fungal DNA barcode, the nrITS region has become the de facto first step in identifying fungi using the phylogenetic species concept [2]. In establishing the nrITS region as the DNA barcode for fungi, prior and subsequent studies evaluated the region’s efficacy for this purpose. In 2008, Nilsson et al. [3] examined within-species (intra-specific) variations within fungi. Using data obtained from the International Nucleotide Sequence Database (INSD), they evaluated satisfactory sequence data from 4185 fungal species to determine intraspecific variations in the entirety of the nrITS sequence data and between the parts of the nrITS (the ITS1 and ITS2 regions). This evaluation showed that 75% or more of species across the kingdom fungi were ≤3% variable in the nrITS region, regardless of the region (whole, ITS1, or ITS2). When Schoch et al. [1] declared the nrITS region to be the appropriate region for the fungal DNA barcode in 2012, they evaluated its efficacy in terms of the relative ease of producing sequence data from it, the availability of existing data, and its ability to recognize species. This later determination was made through the evaluation of the DNA barcode gap in the nrITS and other regions by examining 142 species of Pezizomycotina (Ascomycota) and 43 species of Basidiomycota.

The DNA barcode gap is defined as the region (e.g., gap) that separates the distribution of intra-specific pairwise distances from inter-specific distances among related taxa using a molecular barcode (Figure 1) [4]. Discussions surrounding the barcode gap, its definition and its appropriate application began in the early 2000s at the height of the debate about the efficacy of using DNA barcodes in diversity studies [5]. At this time, the discussion around the behavior and interpretation of the barcode gap was somewhat hypothetical as sufficient sequence data to explore this phenomenon were limited. Current interest in fungal DNA barcodes invites scrutiny toward their application, and barcode gap analysis could be used toward this end [6].

Many studies that have used the fungal DNA barcode to explore anything from fungal taxonomy to the documentation of fungal species richness. Fewer studies have evaluated the barcode gap. This feature in fungal barcodes has been evaluated in Ascomycete groups like *Morchella* [7] and *Cryptococcus* [8]. In 2017, Badotti et al. performed the most comprehensive evaluation of Basidiomycota to date and identified variations in the barcode gap among genera such as *Agaricus*, *Hebeloma* and *Lactarius* [9]. This study relied on the taxonomic identifications supplied by the databases to detect barcode gaps in some genera. A study that produces side-by-side comparisons of the barcode gap among multiple macrofungal genera using expertly curated datasets can yield important insights into the efficacy of the barcode gap for documenting and quantifying fungal diversity.

The evaluation of DNA barcode data in fungi has increased in as high-throughput sequencing technology (e.g., Illumina, 454 pyrosequencing, etc.) has increased the availability of barcode data from environmental samples to quantify and characterize fungal diversity [10]. Most of the early high-throughput technologies produced short-read sequences that were limited to either the ITS1 or ITS2 region of the nrITS. Improvements in sequencing quality and depth allowed for a deep exploration of fungal richness in environmental samples via metabarcoding studies. However, debate surrounding which portion of the nrITS region, ITS1 or ITS2, was better had developed [11,12,13]. Adding to the discussion around metabarcoding studies is that the identification of fungi depends upon adequate representations of nrITS data from accurately identified fungal taxa in sequence databases [14,15,16]. This is one of the many reasons for the GenBank Fungi RefSeq ITS Project (https://www.ncbi.nlm.nih.gov/bioproject/PRJNA177353/).

Increasing the representation of sequences from reference specimens contained within fungaria will continue to improve such databases [17]. Unfortunately, time is not kind to DNA, making it increasingly difficult to obtain a whole DNA barcode from older material [17,18,19]. The adoption of high-throughput sequencing technology for the purpose of DNA barcoding macrofungal collections is a potential solution to the problem of old DNA [18,20,21]. Both Forin et al., 2018, and Miller et al., 2022, demonstrated the efficacy of Illumina MiSeq sequencing in targeting the ITS2 region of old and ancient fungarium specimens [21,22]. This approach has also been demonstrated in lichens [23,24,25]. Adoption of high-throughput sequencing technology has proven to be more successful than Sanger methods in sequencing ancient DNA from specimens and the ability to multiplex hundreds of samples increases the efficiency in terms of time and money [19,22]. The ability to acquire sequence data from numerous, ancient fungarium collections adds to a collection’s value by incorporating more specimens into future systematic and taxonomic studies.

The effectiveness of using either ITS1 or ITS2 to identify taxonomic diversity has been explored in many studies (e.g., [3,9,26,27]). Many of these identify ITS1 as the most effective portion of the nrITS region for species delineation, while some metabarcoding studies found that the ITS2 region is more successful at documenting fungal richness in mixed communities [13,28]. The existence of introns in the 5′ end of the small subunit in certain Ascomycota lineages likely limits primer annealing and makes amplifying the ITS1 region challenging [29,30]. Similarly, the length heterogeneity of the ITS1 region complicates sequencing in genera like *Cantharellus* [31], *Astraeus*, *Russula* and *Lactarius* [32]. These challenges tend to undermine the utility of the ITS1 region. Regardless, the higher rate of variation in ITS1 relative to ITS2 argues in favor of its efficacy in DNA barcoding [26]. How the nrITS, ITS1 and ITS2 regions differ in efficacy across taxa can be explored by evaluating intra- vs. inter-specific pairwise distances. The increased availability of nrITS sequence data, along with the availability of comprehensive taxonomic studies, provides an opportunity for such a study.

Despite technical and physical challenges of obtaining full length nrITS sequence data from ancient specimens, its continued use, either in whole or in parts, will continue to aid taxonomic and systematic studies of many groups of macrofungi (e.g., the fungi which produce macroscopic sexual reproductive structures in the phyla Ascomycota and Basidiomycota—primarily the subphyla Pezizomycotina and Agaricomycotina, respectively). Its application has even expanded to members of the amateur mycological community who have taken up DNA barcoding in order to contribute to fungal identification [33,34,35].

This study aims to explore intra- and inter-specific variations in the nrITS, ITS1 and ITS2 regions of numerous macrofungal lineages to evaluate the barcode gap. This will be accomplished using the pairwise distances of nucleotide sequences and phylogenetic data using datasets that were assembled by taxonomic experts or defined in recent and comprehensive taxonomic studies. The results will help provide perspective on the variation in nrITS DNA barcode sequence data in macrofungi toward developing best practices for fungal taxonomic and environmental research.

## 2. Materials and Methods

Datasets for macrofungal genera representing multiple phyla and orders were assembled according to recent taxonomy and the availability of relevant sequence data. These datasets represent Agaricales (*Hebeloma* [36,37,38,39,40,41,42,43,44,45,46], *Laccaria* [47,48], and *Marasmius*), Boletales (*Suillus* [49]), Russulales (*Russula*, *Stereum* [50]), Polyporales (*Trametes* [51]), Thelephorales (*Sarcodon*), Cantharellales (*Hydnum*), Pezizales (*Morchella*) and Eurotiales (*Elaphomyces* [52,53]) (Figure 2). The assignation of nrITS sequences to species followed the conclusions made by the authors or the conclusions of the referenced taxonomic studies. The author assignation of species to sequences followed taxonomic delineations defined in the taxonomic literature through phylogenetic and morphological characters. These were combined with the instincts of the authors, who have expert knowledge of these groups. All efforts were made to include ≥3 sequences per species to capture intraspecific variation. However, if only one sequence was available for a bona fide species, it was included to assist in measuring interspecific variation.

The genus *Morchella* was split into two datasets based on challenges with aligning nrITS sequence data across the entire genus. The two datasets represent the *M. elata* group and the *M. esculenta* group. The genus *Stereum* was analyzed as a single dataset using two different taxonomic approaches based on the study by DeLong-Duhon in 2020 [50]. One approach is identified as the “lumping” model, which categorized sequences into twenty taxa. The other is the “splitting” model, which categorized sequences into thirty taxa.

For each genus studied, sequence data were assembled in manual alignment editors, Mesquite [54] or AliView [55], with automated alignment performed using Muscle [56] or MAAFT [57]. Matrices were then inspected, and additional adjustments were performed manually to correct misalignments. Sequence data that did not include ≥50% of the ITS1 or ITS2 regions were excluded from the analysis. Aligned matrices were saved into three files: (1) the entire nuclear ribosomal internal transcribed spacer region (ITS1+5.8S+ITS2) with 18S and 28S portions trimmed (nrITS), (2) the ITS1 only, and (3) the ITS2 only. These files were each saved in an aligned fasta format for a pairwise distance analysis. Phylogenies of the nrITS, ITS1, and ITS2 datasets were produced in RAxML [58], using default parameters with a GTR model of evolution, implemented via the CIPRES Web Portal [59], or run on local computers. The results of the bipartition files were saved as newick files. Altogether, six data files were created for each of the eleven genera. These files represent the nrITS, ITS1, and ITS2 datasets in fasta and newick formats.

Analyses of the data from the fasta and newick files were performed in R. A pairwise distance analysis of the aligned nucleotide fasta files was carried out using the dist.dna function from the ape package, with the parameter pairwise.deletion set to TRUE to prevent the removal of sites with missing data (i.e., to keep the characters with gaps) [60]. Analyses were performed with the default K80 distance parameters (different rates of transitions and transversions), as well as the raw parameters to approximate p-distance values. A phylogenetic pairwise distance analysis was performed on the RAxML newick files using cophenetic.phylo() from the ape package, which provided values of the distances between tips.

The distributions of the pairwise distance data were evaluated using boxplots, calculations of the mean, median, variance, and quantiles. A “stress test” of the barcode gap was performed by adjusting the quantiles around the distribution at 85%, 90% and 95% quantiles (Figure 1b). For each of these probabilities the upper quantile of the intra-specific pairwise distance distribution was subtracted from the lower quantile of the inter-specific pairwise distance distribution. A positive value indicates the size of the gap between the intra- and inter-specific distributions. A negative number indicates the absence of a gap. This barcode gap assessment was tested on the pairwise distributions of the nrITS, ITS1 and ITS2 datasets from each genus. Shrinking or loss of the barcode gap was observed as the quantiles around the distributions increased.

## 3. Results

The R markdown files used for analysis, including the data from the aligned fasta and RAxML newick format results, are publicly available through the Open Science Framework (OSF) and on our GitHub repository. For access, please see the Data Availability Statement at the end of the manuscript. The files and the R scripts are annotated and provide access to the data necessary for the absolute reproducibility of the results in this study.

This study assembled twelve datasets representing a total of 5147 sequences and 716 species. These datasets represent individual macrofungal genera from eight orders (Agaricales, Boletales, Russulales, Polyporales, Thelephorales, Cantharellales, Pezizales and Eurotiales) and two phyla (Ascomycota and Basidiomycota). The number of species per dataset consisted of as few as 11 (*Sarcodon*) to as many as 148 (*Russula*), with the number of sequences ranging from 62 (*Sarcodon*) to 1616 (*Hebeloma*). The nrITS datasets ranged from 656 (*Trametes*) to 1356 (*Morchella esculenta* clade) nucleotides in length, the ITS1 datasets ranged from 183 (*Stereum*) to 756 (*Morchella esculenta* clade) nucleotides in length and the ITS2 datasets ranged from 235 (*Stereum*) to 455 (*Morchella esculenta* clade) nucleotides in length. The mean aligned lengths of the datasets were 820.1 (nrITS), 333.3 (ITS1) and 311.7 (ITS2) nucleotides.

The results for the pairwise distance analyses using the K80 and raw models and for the phylogenetic pairwise distances are comparatively similar. Because of this, most of the data presented below reflect the analyses using the K80 model unless otherwise specified. The results for the raw model and phylogenetic pairwise distance analyses are presented in the Appendix A.

The results of the pairwise distance analysis of the sequence data are summarized in Figure 3 as boxplots. The overall means from 5147 sequences comparing the intra-specific nrITS, ITS1, and ITS2 pairwise distance results are 0.006, 0.013 and 0.008 respectively. The inter-specific pairwise distance means are 0.061, 0.099 and 0.079, which are displayed as horizontal dotted lines for the nrITS (black), ITS1 (red) and ITS2 (gray) datasets in Figure 3. These mean values are likely skewed by the *Hebeloma* data from this genus comprise around one-third of the entire dataset. The mean differences in the sequence data between the inter- and intra-specific distances for nrITS, ITS1 and ITS2 are 0.055, 0.087 and 0.071, respectively. Similar results are demonstrated when the pairwise distances were analyzed using a “raw” model of evolution (Appendix A) or phylogenetic distances (Appendix A).

The variances in the intra-specific pairwise distance distributions for these regions are 0.0001, 0.0004 and 0.0002, with the ratio of ITS1 to ITS2 nearly double at 1.93. The inter-specific distribution variances were 0.0032, 0.0112 and 0.0071, with an ITS1 to ITS2 ratio of 1.57. The ITS1 region had a higher mean and more variation than the ITS2 region in both inter and intra-specific pairwise distances of the sequence data given the above ratios. This is visually observed in Figure 3 and Figure 4. The latter presents a bar graph that displays the variances in pairwise distances, analyzed using the “raw” model, for macrofungal genera.

The barcode gap is defined as the difference between the highest intra-specific and lowest inter-specific distance measurements. This is what is expected when using a dataset that does not contain aberrations or outliers in the data. However, in most datasets in this study, the lower limits of the inter-specific pairwise distances contained a zero, negating any evaluation of the barcode gap. The barcode gap “stress test” presented in Table 1 was performed to address this. The expectation was that the quantiles will drop the outlying tails of the intra- and inter-specific distributions and provide an opportunity to evaluate the barcode gap (Figure 1b).

In summary, when assessing the barcode gap using quantiles, the entire nrITS region and the ITS2 region produce similar results at the 85, 90 and 95% quantiles using distance analyses under the K80 model (Table 1). These datasets retained their barcode gaps at the 90 and 95% quantiles more often in macrofungal taxa relative to the ITS1 datasets. The entire nrITS and ITS2 region produced the most consistent barcode gaps across the studied datasets. The ITS1 region resolved the fewest number of barcode gaps among the sequence data.

At the 85% quantile, the nrITS region produced a barcode gap for the majority of the datasets (11/13). The exceptions to this are the *Trametes* and the *Morchella elata* groups (avg. size size among existing barcode gaps: 0.0278, Table 1). The results were similar when assessing the 85% quantile for the pairwise distance of ITS2, in which the *Stereum* “lumping” and *Morchella elata* groups were the only datasets lacking a barcode gap (avg. size size among existing barcode gaps: 0.0320). For the 85% quantile for the pairwise distance of the ITS1 region, only 9 of the 13 datasets had sufficient barcode gaps. The *Stereum* “lumping” and “splitting”, *Trametes* and *Morchella elata* group were the datasets where a barcode gap was not observed (avg. size size among existing barcode gaps: 0.0350). At the 90% quantile, a majority of the nrITS region (10/13) and the ITS2 region (8/13) were retained. At the 95% quantile, these two datasets retained slightly less than half of the barcode gaps across all taxa (6/13). In contrast, the ITS1 region retained gaps in only four datasets (the *Russula, Trametes, Sarcodon* and *Morchella esculenta* groups) at both the 90% and 95% quantiles.

The variances in intra- and inter-specific pairwise distance distributions were evaluated across sequences and taxa. Under all models evaluated, *Marasmius* has the most intra-specific variation, while *Hydnum* has the lowest (Figure 3). For inter-specific variation, *Elaphomyces* has highest while the *Stereum* “lumping” model has the lowest. Figure 3 compares intra-specific and inter-specific variances for datasets across nrITS, ITS1 and ITS2 data. All distribution results of means and variances under K80, raw, and phylogenetic analysis can be found in Appendix A.

The genus *Stereum* was analyzed using two different approaches to species recognition, “lumping” and “splitting”, according to DeLong-Duhon et al. (unpublished but available on bioRχiv [50]). Their study applied Assemble Species by Automatic Partitioning (ASAP) method using MAFFT [57], and the species partitioning with the lowest (e.g., best) scores estimated either 20 or 30 species as optimal partitions. A pairwise distance analysis of *Stereum* was performed with both these approaches, and similar results were produced using the K80 and raw models and using phylogenetic distance data. In Figure 3, the lumping partitioning results are displayed in colored outlines and white-filled boxplots, while the splitting partioning results are presented as color-filled boxplots. Each result is displayed side by side for intra- and inter-specific nrITS, ITS1 and ITS2 comparisons. In Figure 4, *Stereum* under the lumping model (*Stereum_20*) has the second-highest variance for intra-specific pairwise distributions (Figure 4a). By comparison the splitting model (*Stereum_30*) is number eight on this list of intra-specific variances. In the inter-specific comparisons this relationship is inversed where the lumping model has the lowest variance and the splitting model has the second-lowest variance (Figure 4b).

## 4. Discussion

Using the barcode gap to analyze nrITS sequence data across macrofungal genera produced a number of interesting observations. First, variance in distance scores appears to have the greatest influence on the presence of a barcode gap. This can be seen in comparisons of nrITS, ITS1, and ITS2 sequence data, along with approaches used to recognize species diversity. Secondly, using the barcode gap to identify cutoffs for sequence clustering is challenging. One reason for this is that barcode gaps differ widely across taxonomic groups. The other reason is the fact that intra- and inter-specific distributions often overlap, eliminating the ability to use the barcode gap to infer cutoffs. This supports the discussion that nrITS sequence data has limited efficacy for taxonomic identification and verification across macrofungal genera [2].

ITS1 vs. ITS2–The ITS1 sequence data alone are nearly twice as variable as that of the ITS2 data (Figure 4). This difference between the two regions has been widely observed in studies that evaluate the DNA barcode in fungi [11,12,13]. It is also correlated with fewer observations of barcode gaps from the ITS1 datasets in the barcode gap assessment (Table 1). In the *Laccaria*, *Russula*, *Suillus*, *Sarcodon*, *Hydnum* and *Stereum* “splitting” datasets a barcode gap was retained for ITS2 all the way up through the 95% quantiles. In contrast, only four ITS1 datasets were retained at the 95% quantile. One difference being that the *Morchella esculenta* group was retained for ITS1 and not ITS2 at this quantile. Of all the taxa, *Marasmius* has the highest intra-specific variance. It retained a barcode gap for all sequence data at the 85% quantile and for the nrITS at the 90% quantile. For all other comparisions the barcode gap was lost (Table 1).

The fact that the ITS1 region struggles to maintain its barcode gap should affect the interpretation of ITS1′s role as a superior barcode marker [26]. Regardless, even though the presence or absence of a barcode gap can affect species recognition, it is still a theoretical approach to assessing species. It is not the final determination for species recognition. What these results do reinforce is the need to gather additional data in order corroborate evidence of species in taxonomic studies [61]. For environmental metabarcoding studies, it would be useful to broadly screen multiple taxonomic groups to understand the relative thresholds for recognizing MOTUs from the environment. This is discussed more below.

“Lumping” vs. “Splitting”–How species are partitioned within a taxonomic group affects the variance in pairwise distance measurements. As expected, this also influences the barcode gap. The *Stereum* dataset, was analyzed using “lumping” and “splitting” approaches in which the same dataset was divided into 20 species or as many as 30 species, respectively. Figure 5 illustrates how intra-specific variance is broader in the “lumping” hypothesis and narrower in the “splitting” hypothesis (Figure 5a,b respectively). In contrast, inter-specific variation is only slightly narrower for the “lumping” approach than the “splitting” (Figure 5c,d). When looking at the barcode gap stress test of *Stereum*, the wider variance in intra-specific variation in the “lumping” model appears to have a greater affect than the widening of the inter-specific variance in the “splitting” model.

Variance values for all K80 analyses are presented in Appendix A (raw and phylogenetic results are in Appendix A respectively). In comparing the variances for *Stereum* for the “lumping” and “splitting” approaches, the intra-specific variances are 1.54 × 10^−4^ and 4.52 × 10^−5^ respectively. The intra-specific variances are 1.75 × 10^−4^ and 2.04 × 10^−4^. If we take the high/low variance ratio between “lumping” and “splitting”, the ratio for intra-specific variances is near 3.5, while for inter-specific variances it is 1.2. This demonstrates how intra-specific variance is influenced under the “lumping” and “splitting” approaches with *Stereum*. Whether this observation is universal needs to be evaluated further in additional taxa.

Estimating cutoffs for MOTUs using barcode gaps–The ideal cutoff for recognizing MOTUs in environmental sequence data has been evaluated [3], but the challenge is in applying such universal cutoffs to a pool of sequences that likely represent fungi from multiple phyla [62]. Table 2 shows that establishing cutoffs for the genera in this study using the barcode gap is challenging, even for expertly curated datasets. Regardless, the fact that the middle of the proposed barcode gap for the taxa in this study varies from less than 2% to nearly 6% shows that a universal cutoff runs the risk of over- or underestimating fungal diversity, depending on the taxa.

These results are precisely what was described by Ryberg (2015) when looking at the misrepresentation of species when assessing MOTUs [62]. He pointed out that several studies have observed an overlap of within- (intra-) and between-species (inter-) distances. The reality is that this overlap eliminates the barcode gap in nrITS sequence data, severely challenging the use of the marker for species recognition.

While the risk of over- or underestimating species diversity in environmental studies is real, the efforts needed to address these issues are likely to be impractical. Understanding community diversity and composition can be an abstract exercise. To explore how fungal communities contribute to ecosystem health, understanding the approximate diversity of a taxonomic group is sufficient if you have an understanding of the functional roles of the group. In such cases, working with sequence similarity cutoffs of 95-98% in forming clusters for MOTUs ought to provide researchers with sufficient resolution to characterize fungal communities.

Applying nrITS for taxonomy–Regarding describing new species, nrITS or its parts can aid in identifying novel taxa, but multiple molecular markers are necessary and strongly advised when delineating new species [61]. Fungal species are more than their sequence data. As a result, the current recommendation is refrain from using under-complex methods for species identification and to use multi-dimensional approaches such as the “consolidated species concept” [63]. Approaches that combine morphological, ecological and phylogenetic species concepts, using evidence from multiple molecular markers, are critical for achieving an accurate interpretation of taxonomic diversity [64]. Studies that describe new species of fungi would be of greater value if they provided a proper exploration of the fungal barcode gap by defining intra- and inter-specific distances. This would improve the utility of nrITS sequence data in the future.

Applying nrITS for species identification–This study evaluated the variance in nrITS sequence data across macrofungi because there is a growing need to produce DNA sequence data that improves reference sequences in databases [65,66]. Two approaches, which are not mutually exclusive, can be made toward this end: (1) increasing the representation of sequence data from curated fungarium specimens (quantity) and (2) increasing the expert evaluation of existing and new sequence data (quality). Ideally, reference sequences would be vastly improved by both, but in many cases, it has been the lack of quality, not quantity, that limits the usefulness of reference databases. In 2019, Hofstetter et al. highlighted the difficulty of using databases like GenBank nucleotide (nt) due to 30% of the sequences being erroneous in their identifications [14]. Olds et al., 2023, likewise estimated that 65% of the fungal specimens in global collections are misidentified, have outdated identifications, or have not been identified [19]. Additionally, metabarcoding studies frequently encounter dark taxa from environmental samples which are potentially a symptom of missing representation of known fungal taxa in existing sequence databases [67,68,69]. These dark taxa are even encountered in fungaria [70]. To address this, some researchers request that only sequence data from types be used [71]. However, as mentioned previously, the DNA from type specimens might not be obtainable, and the type sequence alone does not capture intra-specific variation for the species. The ability to provide quality reference data requires the preservation of source specimens from which additional taxonomic data can be evaluated. This emphasizes the importance of sequencing macrofungal collections to enable the recognition and identification of fungal taxa.

Macrofungi are a low-hanging fruit with respect to applying the fungal DNA barcode because these fungi produce sporocarps that are easily vouchered, referenced, and sequenced relative to microfungi and dark fungal taxa (DFT). For the latter, the calls to utilize sequence data as the type reference for DFT have increased in the last decade [72,73,74]. In 2023, Nilsson et al. revised the debate by establishing a set of best practices for defining species from DNA sequence data only [74]. While the current study addresses macrofungi, it provides some context for the discussion of DFT in the need to understand the boundaries around intra-specific sequence variation when establishing species. This would speak to the reproducibility of and confidence in recognizing taxa in future sequencing efforts.

The continued application of the barcode gap to fungi–The DNA barcode gap offers insight into the efficacy of nrITS sequence data or any additional barcode sequence data in macrofungi. Based on the work of this and other studies, the application of the barcode gap in fungi has its limitations [9,62]. Phillips et al., 2022, have cautioned that too many studies apply the DNA barcode to their organisms without sufficient statistical scrutiny of the barcode gap [6]. While they do provide strong arguments for scrutinizing the barcode gap, in fungi this may be moot where the barcode gap does not exist using nrITS sequence data. The definition of a fungal species is governed by so much more than a single molecular region. As a result, scrutinizing barcode gaps in secondary and tertiary barcodes for fungi is a practical next step in order to satisfy the desire of using molecular data for species recognition and identification.

In this study we have demonstrated that several macrofungal groups lack a clear barcode gap. This reality likely stems from wide variances in their intra- and inter-specific distance distributions. Compounding the issue is how variable distance measurements can be between taxa and between nrITS, ITS1 and ITS2 regions. If seeking a barcode gap signal from sequence data, the ITS2 region was more reliable than the ITS1 region across the taxa used in this study. This study adds to the already significant wealth of information about the strengths and weaknesses of the fungal DNA barcode and the growing emphasis of using additional molecular markers in taxonomic and systematic studies of fungi.

## Figures and Tables

**Figure 1 jof-09-00788-f001:**
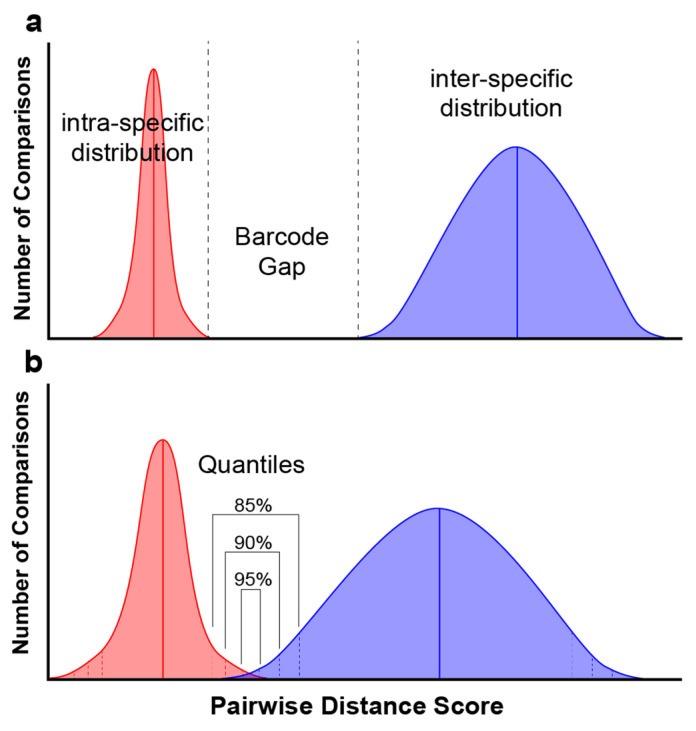
Distributions of intra- and inter-specific pairwise distances around the barcode gap (**a**). When means are closer and/or variances around the means are greater, distributions can overlap, eliminating the barcode gap (**b**). An examination of distribution quantiles can be used to understand the effectiveness of molecular barcode sequence data and the barcode gap for a taxon.

**Figure 2 jof-09-00788-f002:**
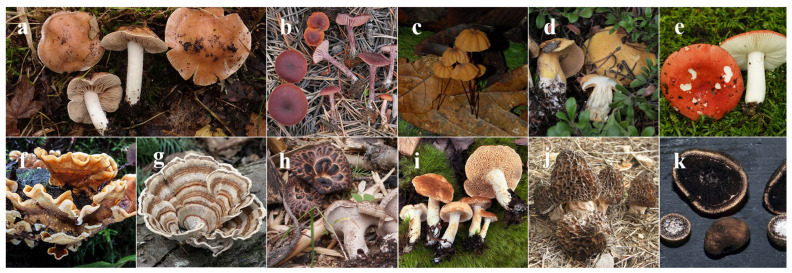
The following macrofungal genera were studied: (**a**) *Hebeloma celatum* (H. J. Beker), (**b**) *Laccaria amethystina* group, (**c**) *Marasmius* sp., (**d**) *Suillus tomentosus* (N. Nguyen), (**e**) *Russula magnarosea* clade (C. R. Noffsinger), (**f**) *Stereum hirsutum* (M. G. Wood, Mycoweb.org), (**g**) *Trametes versicolor* (M. G. Wood), (**h**) *Sarcodon imbricatus*, (**i**) *Hydnum* sp. (R. A. Swenie), (**j**) *Morchella elata* group and (**k**) *Elaphomyces muricatus* (M. G. Wood). All pictures by A. W. Wilson unless otherwise indicated.

**Figure 3 jof-09-00788-f003:**
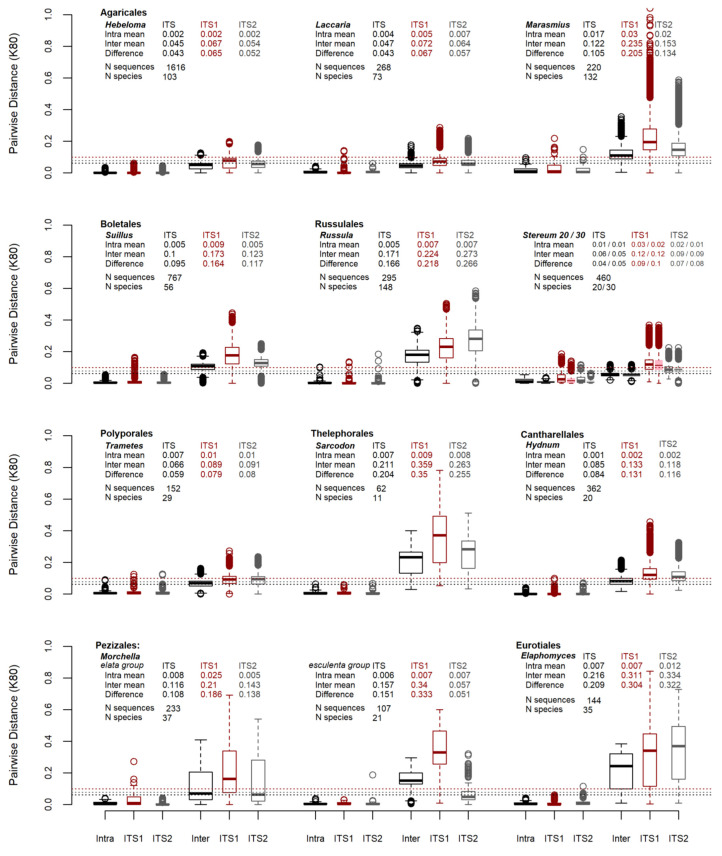
Pairwise distances of DNA barcode sequence data using a K80 substitution model. Horizontal lines are the inter-specific means for the nrITS, ITS1 and ITS2 pairwise distance distributions from 5146 sequences. *Stereum* is evaluated under two partitions. The “lumping” partition has 20 species and is represented by open boxplots. The “splitting” partition has 30 species and is represented in the filled boxplots.

**Figure 4 jof-09-00788-f004:**
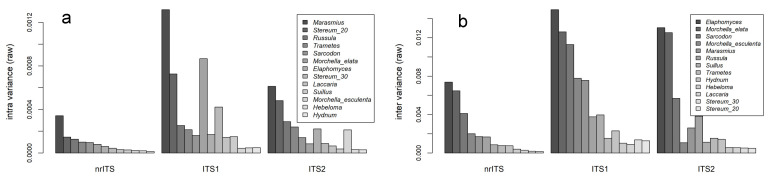
Variances in pairwise distance distributions, showing ranked nrITS sequence data for intra-specific (**a**) and inter-specific (**b**) distances.

**Figure 5 jof-09-00788-f005:**
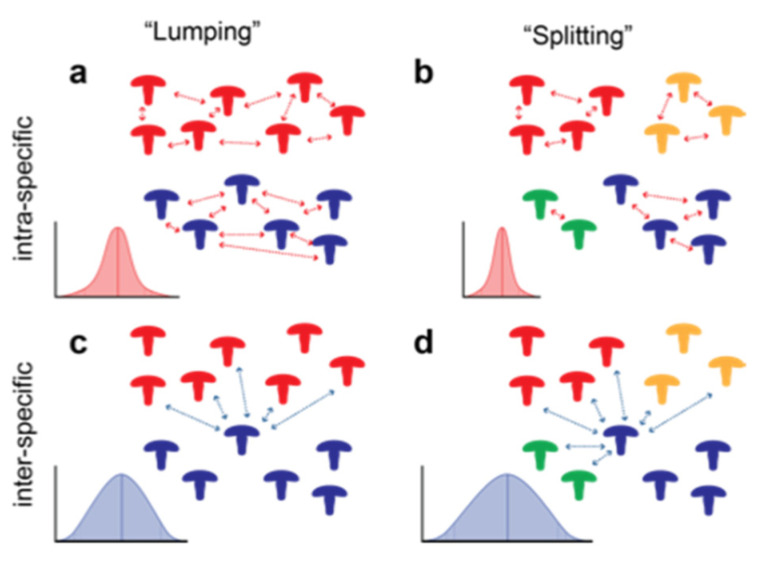
How “lumping” and “splitting” approaches to species delineation affect pairwise distance distributions. Intra-specific variation is broader when using the “lumping” approach (**a**, red distribution) than when using the “splitting” approach (**b**). For inter-specific variation, this relationship is inverted such that “lumping” (**c**, blue distribution) is narrower than “splitting” (**d**).

**Table 1 jof-09-00788-t001:** Barcode gap “Stress Test”: assessment of the difference between minimum inter- and maximum intra-specific pairwise distance values at progressive quantiles of distance distributions. (K80 model data).

			85% Quantiles						90% Quantiles						95% Quantiles					
			nrITS		ITS1		ITS2		nrITS		ITS1		ITS2		nrITS		ITS1		ITS2	
Name	Ntaxa	Nseq	Gap?	Size	Gap?	Size	Gap?	Size	Gap?	Size	Gap?	Size	Gap?	Size	Gap?	Size	Gap?	Size	Gap?	Size
** *Hebeloma* **	103	1616	**TRUE**	0.0051	**TRUE**	0.0043	**TRUE**	0.0052	**TRUE**	0.0016	FALSE		**TRUE**	0.0000	FALSE		FALSE		FALSE	
** *Laccaria* **	73	268	**TRUE**	0.0121	**TRUE**	0.0094	**TRUE**	0.0105	**TRUE**	0.0063	FALSE		**TRUE**	0.0100	**TRUE**	0.0018	FALSE		**TRUE**	0.0000
** *Marasmius* **	132	220	**TRUE**	0.0152	**TRUE**	0.0128	**TRUE**	0.0046	**TRUE**	0.0075	FALSE		FALSE		FALSE		FALSE		FALSE	
** *Suillus* **	56	767	**TRUE**	0.0230	**TRUE**	0.0153	**TRUE**	0.0331	**TRUE**	0.0142	FALSE		**TRUE**	0.0235	**TRUE**	0.0013	FALSE		**TRUE**	0.0039
** *Russula* **	148	295	**TRUE**	0.0771	**TRUE**	0.0720	**TRUE**	0.1192	**TRUE**	0.0648	**TRUE**	0.0531	**TRUE**	0.0900	**TRUE**	0.0380	**TRUE**	0.0274	**TRUE**	0.0515
** *Stereum* **																				
“lumping”	20	460	**TRUE**	0.0036	FALSE		FALSE		FALSE		FALSE		FALSE		FALSE		FALSE			
“splitting”	30	460	**TRUE**	0.0156	FALSE		**TRUE**	0.0322	**TRUE**	0.0114	FALSE		**TRUE**	0.0230	**TRUE**	0.0058	FALSE		**TRUE**	0.0125
** *Trametes* **	29	152	FALSE		FALSE		**TRUE**	1E-04	FALSE		FALSE		FALSE		FALSE		FALSE		FALSE	
** *Sarcodon* **	11	62	**TRUE**	0.0413	**TRUE**	0.0472	**TRUE**	0.0527	**TRUE**	0.0152	**TRUE**	0.0423	**TRUE**	0.0134	FALSE		**TRUE**	0.0277	**TRUE**	0.0000
** *Hydnum* **	20	362	**TRUE**	0.0425	**TRUE**	0.0552	**TRUE**	0.0503	**TRUE**	0.0349	**TRUE**	0.0418	**TRUE**	0.0450	**TRUE**	0.0267	**TRUE**	0.0258	**TRUE**	0.0313
** *Morchella* **																				
*elata* group	37	233	FALSE		FALSE		FALSE		FALSE		FALSE		FALSE		FALSE		FALSE		FALSE	
*esculenta* group	21	107	**TRUE**	0.0306	**TRUE**	0.0607	**TRUE**	0.0043	**TRUE**	0.0196	**TRUE**	0.0407	FALSE		**TRUE**	0.0029	**TRUE**	0.0082	FALSE	
** *Elaphomyces* **	35	144	**TRUE**	0.0397	**TRUE**	0.0380	**TRUE**	0.0393	**TRUE**	0.0103	FALSE		**TRUE**	0.0199	FALSE		FALSE		FALSE	
			**Count**	**AVE**	**Count**	**AVE**	**Count**	**AVE**	**Count**	**AVE**	**Count**	**AVE**	**Count**	**AVE**	**Count**	**AVE**	**Count**	**AVE**	**Count**	**AVE**
			11	0.0278	9	0.0350	11	0.0320	10	0.0186	4	0.0445	8	0.0281	6	0.0127	4	0.0223	6	0.0166

**Table 2 jof-09-00788-t002:** Barcode gap presence and mean values estimated at 95% quantiles from p-distance (raw) data.

	ITS		ITS1		ITS2	
Names	Gap?	Mean	Gap?	Mean	gap?	Mean
** *Hebeloma* **	FALSE	0.7%	FALSE	1.1%	FALSE	1.2%
** *Laccaria* **	**TRUE**	1.8%	FALSE	2.5%	FALSE	2.5%
** *Marasmius* **	FALSE	5.0%	FALSE	10.1%	FALSE	5.9%
** *Suillus* **	**TRUE**	1.7%	FALSE	2.9%	**TRUE**	2.1%
** *Russula* **	**TRUE**	4.0%	**TRUE**	4.5%	**TRUE**	5.7%
** *Stereum* **						
“lumping”	FALSE	3.5%	FALSE	6.9%	FALSE	5.4%
“splitting”	**TRUE**	2.5%	FALSE	5.8%	**TRUE**	3.3%
** *Trametes* **	FALSE	1.8%	FALSE	2.4%	FALSE	2.9%
** *Sarcodon* **	FALSE	3.7%	**TRUE**	5.5%	**TRUE**	3.7%
** *Hydnum* **	**TRUE**	2.0%	**TRUE**	2.8%	**TRUE**	2.9%
** *Morchella* **						
elata group	FALSE	1.8%	FALSE	5.0%	FALSE	1.8%
esculenta group	**TRUE**	1.6%	**TRUE**	2.4%	FALSE	1.1%
** *Elaphomyces* **	FALSE	2.4%	FALSE	4.0%	FALSE	4.2%
Count	6		4		5	
Max		4.0%		5.5%		5.7%
Min		1.6%		2.4%		2.1%

## Data Availability

Data and scripts used in this study can be found through the OSF link: https://osf.io/rvk42/; and through the GitHub repository: https://github.com/DenverBotanicGardens/DNA-Barcode-Gap-Project.

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
