# Peer review of "Does One Size Fit All? Variations in the DNA Barcode Gaps of Macrofungal Genera"

_jof, 2023, doi:10.3390/jof9080788_

Round 1
Reviewer 1 Report
All my comments and suggestions concerning the text are included as comments in the pdf.

Author Response
All the comments and suggestions made by this reviewer are in an attached PDF.
- We have incorporated all of the changes this reviewer has suggested for the Introduction, Materials and Methods, and Results sections.
- For the Discussion we have re-written the first paragraph as suggested by the reviewer.
- In addition to this, the Discussion section has been further revised to incorporate observations and epiphanies regarding the utility of the barcode gap and nrITS sequence data.
Reviewer 2 Report
This study evaluates the barcoding gap and molecular identification in 11 fungal genera. Some of the fungi evaluated form mycorrhizal fungi, a field in which molecular identifications and the use of metabarcoding have become indispensable in recent years. I therefore consider the study interesting, but it is a bit poor, as it does not explore in depth the problems of molecular identifications in the selected genera. It is not only the absence of barcoding gap that causes problems in the identifications.
Since the authors do not generate new data, they use data published by other authors, they should make a little more effort in analyzing the data and explore in depth the molecular identifications of the selected groups. In the discussion the authors provide data (based on references) for identification errors using databases such as GenBank. However, they do not make this assessment for the selected genera. In addition to the barcoding gap, identification success should be evaluated using the BLAST tool. It would also be desirable for the authors to evaluate and discuss threshold values for generating OTUs in the study genera or conversely the use of ASV for metabarcoding identifications. This issue can do relatively easily based on genetic distances calculated in the study.
In the discussion the authors talk about barcoding being a first step in species delimitation. But the goal of barcoding analyses is not species delimitation which should be done by taxonomists using various gene regions in combination with phenotypic, ecological and distributional data.
Since I consider that the manuscript needs a major revision before publication, I have not made an exhaustive revision of the text, although several parts need substantial improvement. Some comments are listed below.
Line 48: “INSD” please indicate what the abbreviations mean
Lines 87-89: I don't know if I agree with this statement. NGS technologies have increased the number of ITS sequences but there is really a lack of evaluation of the existence of barcoding bap as well as the success of successful identifications using BLAST. For this reason, the authors' work is interesting.
Line 105: “Both Forin et al. 2018 and Miller et al. 2022…” There are few more references:
Kistenich, S., Halvorsen, R., Schrøder-Nielsen, A., Thorbek, L., Timdal, E., & Bendiksby, M. (2019). DNA sequencing historical lichen specimens. Frontiers in Ecology and Evolution, 7, 5.
Gueidan, C., & Li, L. (2022). A long-read amplicon approach to scaling up the metabarcoding of lichen herbarium specimens. MycoKeys, 86, 195.
Leavitt, S. D., Kueler, R., Newberry, C. C., Rosentreter, R., & St Clair, L. L. (2019). Shotgun sequencing decades-old lichen specimens to resolve phylogenomic placement of type material. Plant and Fungal Systematics.
Line 158-159: Please, elaborate more on what this sentence means
Line 163-164: More than 3 sequences is too ambiguous. Indicate the mean number of sequences per species and the range. It is not the same to analyze 3 sequences for species with restricted distributions as for species with wide distributions.
Line 194: The choice of K80 to calculate genetic distances must be justified, since some authors have demonstrated the influence of different distance calculations in barcoding analyses, while others recommend the use of p-distances (Collins et al. 2012).
Line 217-227: I suggest moving this paragraph to material and methods
Lines 228-234: This is methodology
Lines 237-238: Please, indicate the standard desviation for the distances values
Lines 239-240: Move this sentence to the caption of figure 3
Lines 304: What does this mean? does it mean that there is not a barcoding gap with 85% of threshold? Please specify
Lines 419-423: Please, do not repeat the values of genetic distances.
Author Response
We appreciate the comments and critiques of reviewer 2 and have made a good faith attempt the accommodate many of the suggestions they put forth. The current version of the manuscript includes additional analysis and contribute to a significantly improved version of the paper.
“This study evaluates the barcoding gap and molecular identification in 11 fungal genera. Some of the fungi evaluated form mycorrhizal fungi, a field in which molecular identifications and the use of metabarcoding have become indispensable in recent years. I therefore consider the study interesting, but it is a bit poor, as it does not explore in depth the problems of molecular identifications in the selected genera. It is not only the absence of barcoding gap that causes problems in the identifications.”
Additional analysis and rewrites to the Discussion section provides additional context to the study which hopefully satisfies the reviewer. As a meta-study, there are numerous avenues of inquiry that could be explored, but it is necessary to limit the questions in an effort to provide as clear a narrative and study as possible. What constitutes an “in-depth” exploration of the data comes across as vague. We hope the reviewer appreciates the current depths to which we investigated the data.
“Since the authors do not generate new data, they use data published by other authors, they should make a little more effort in analyzing the data and explore in depth the molecular identifications of the selected groups. In the discussion the authors provide data (based on references) for identification errors using databases such as GenBank. However, they do not make this assessment for the selected genera. In addition to the barcoding gap, identification success should be evaluated using the BLAST tool. It would also be desirable for the authors to evaluate and discuss threshold values for generating OTUs in the study genera or conversely the use of ASV for metabarcoding identifications. This issue can do relatively easily based on genetic distances calculated in the study.”
There are several points in this paragraph that we will attempt to address in order.
- “No new data” critique – This is considered a meta-analysis that uses existing data to explore biological questions. New data need not be generated for scientific research. The effort to accumulate and evaluate 1000’s of sequences, assemble and align datasets, and coordinating the analysis to which results can be used to draw conclusions/make observations is significant. Regardless, we have added new figures that demonstrate differences among the taxa in this study with regard to variance as well as illustrations that help explain how use of the barcode gap can affect interpretation of diversity at the species level.
- “identification errors using databases”- It’s unclear which part of the Discussion the Reviewer is referring to. Regardless, the Discussion section has been re-written significantly so this concern may now be moot.
- Use of BLAST – BLAST is a tool for finding sequences in a database that match your query. That’s not what this study is attempting to explore.
- OTUs and ASVs – We address these issues in the new version of the discussion.
“In the discussion the authors talk about barcoding being a first step in species delimitation. But the goal of barcoding analyses is not species delimitation which should be done by taxonomists using various gene regions in combination with phenotypic, ecological and distributional data.”
The new version of the Discussion addresses these concerns.
“Since I consider that the manuscript needs a major revision before publication, I have not made an exhaustive revision of the text, although several parts need substantial improvement. Some comments are listed below.”
We thank the reviewer for the efforts made to review this study and hope that the attempts to address their comments are mostly, if not entirely, satisfactory.
Line 48: “INSD” please indicate what the abbreviations mean - DONE
Lines 87-89: I don't know if I agree with this statement. NGS technologies have increased the number of ITS sequences but there is really a lack of evaluation of the existence of barcoding bap as well as the success of successful identifications using BLAST. For this reason, the authors' work is interesting.
Line 105: “Both Forin et al. 2018 and Miller et al. 2022…” There are few more references:
Kistenich, S., Halvorsen, R., Schrøder-Nielsen, A., Thorbek, L., Timdal, E., & Bendiksby, M. (2019). DNA sequencing historical lichen specimens. Frontiers in Ecology and Evolution, 7, 5.
Gueidan, C., & Li, L. (2022). A long-read amplicon approach to scaling up the metabarcoding of lichen herbarium specimens. MycoKeys, 86, 195.
Leavitt, S. D., Kueler, R., Newberry, C. C., Rosentreter, R., & St Clair, L. L. (2019). Shotgun sequencing decades-old lichen specimens to resolve phylogenomic placement of type material. Plant and Fungal Systematics.
- These have been incorporated into the manuscript.
Line 158-159: Please, elaborate more on what this sentence means
- A critical step to performing this study is how sequence data are assigned to species. The crux of this study evaluates intra- and inter-specific variation for pairwise distance analysis so accuracy in assigning species names to sequences is paramount. Hopefully this makes sense. The sentence in question appears straightforward. I’m not sure the sentence needs to be rewritten given this added context.
Line 163-164: More than 3 sequences is too ambiguous. Indicate the mean number of sequences per species and the range. It is not the same to analyze 3 sequences for species with restricted distributions as for species with wide distributions.
- It is not clear what the justification of “distributions” has on this particular study. I would imagine the reviewer is interested in biogeographic implications of sequence variation. That is a separate question however. As for providing other statistics, we clearly provide the total number of sequences per dataset along with the total number of species in each dataset in Table 1. Readers should be able to infer the mean from this. We also provide access to each of our datasets, scripts, and any other materials readers would like to have access to. If this information such as range is important, it can be referenced.
Line 194: The choice of K80 to calculate genetic distances must be justified, since some authors have demonstrated the influence of different distance calculations in barcoding analyses, while others recommend the use of p-distances (Collins et al. 2012).
- Oddly, the dna.dist() from ape doesn’t have “p-distance”. It does have a “raw” model which we infer to be the same thing. We have incorporated analysis and results under the “raw” model into this study.
Line 217-227: I suggest moving this paragraph to material and methods
- Done
Lines 228-234: This is methodology
- Done
Lines 237-238: Please, indicate the standard desviation for the distances values
- We have provided additional supplemental information that includes the means and variances of all distance values for all analysis for all taxa.
Lines 239-240: Move this sentence to the caption of figure 3
- A sentence describing the horizontal dotted line has been added to the caption of Figure 3.
Lines 304: What does this mean? does it mean that there is not a barcoding gap with 85% of threshold? Please specify
- The TRUE and FALSE statements of Table 1 indicate whether there is a barcode gap between intra- and inter-specific pairwise distance distributions at the quantile and molecular marker specified. At the 85% quantile, nine of the thirteen ITS1 datasets have a barcode gap.
Lines 419-423: Please, do not repeat the values of genetic distances.
- Done.